# Characterising Interventions in Causal Games

**Manuj Mishra**[1]      **James Fox**[1,2]      **Michael Wooldridge**[1]

[1]Department of Computer Science, University of Oxford, UK
[2]London Initiative for Safe AI, UK

## Abstract

*Causal games* are probabilistic graphical models that enable causal queries to be answered in multi-agent settings. They extend causal Bayesian networks by specifying decision and utility variables to represent the agents' degrees of freedom and objectives. In multi-agent settings, whether each agent decides on their policy before or after knowing the causal intervention is important as this affects whether they can respond to the intervention by adapting their policy. Consequently, previous work in causal games imposed chronological constraints on permissible interventions. We relax this by outlining a sound and complete set of primitive causal interventions so the effect of any arbitrarily complex interventional query can be studied in multi-agent settings. We also demonstrate applications to the design of safe AI systems by considering causal mechanism design and commitment.

## 1 INTRODUCTION

When designing a system for rational, self-interested agents, it is important to incentivise behaviour that aligns with high-level goals, such as maximising social welfare or minimising the harm to other agents. To address this, game theory provides several representations that have different strengths and weaknesses depending on the setting. Hammond et al. [2023] recently introduced *causal games* to extend Pearl [2009]'s 'causal hierarchy' to the multi-agent setting. Causal games are graphical representations of dynamic non-cooperative games, which can be more compact and expressive than extensive-form games [Koller and Milch, 2003, Hammond et al., 2021]. Like causal Bayesian networks, they use a directed acyclic graph (DAG) to represent causal relationships between random variables, but they also specify decision and utility variables. Each agent selects a

policy – independent conditional probability distributions (CPDs) over actions for each of their decision variables – to maximise their expected utility.

Causal Bayesian networks handle interventions in settings without agents by cutting any edges incident to the intervened node in the DAG to represent that the effect of an intervention can only propagate downstream. However, to handle how an agent might or might not adapt their policy in response to an intervention, mechanised graphs extend the regular DAG by explicitly representing each variable's distribution and showing which other variables' distributions matter to an agent optimising a particular decision rule [Hammond et al., 2023, Dawid, 2002].

**Related Work:** The effect of causal interventions is important in many fields such as economics [Heckman and Pinto, 2022, LeRoy, 2004], computer science [Brand et al., 2023] and public health [Ahern et al., 2009, Glass et al., 2013]. However, these fields use models that do not account for the strategic nature of multi-agent systems. Recently, causal games [Hammond et al., 2023] were introduced to unify the power of causal and strategic reasoning in one model. Causal games and their single-agent variant, causal influence diagrams [Everitt et al., 2021a], have been used to design safe and fair AI systems [Ashurst et al., 2022, Everitt et al., 2021b, Farquhar et al., 2022, Carroll et al., 2023], explore reasoning patterns and deception [Pfeffer and Gal, 2007, Ward et al., 2022], and identify agents from data [Kenton et al., 2023]. The key limitation is that existing work on multi-agent causal models assumes that an intervention is either fully post-policy (entirely invisible) to all agents or fully pre-policy (entirely visible) to all agents before they decide on their decision rule at each decision point.

**Contributions:** Our most important novel contribution is to extend the theory of interventions in causal games to be able to accommodate arbitrary queries where agents choose their decision rules based on any subset of the interventions (those visible to them). This is necessary to discuss richer

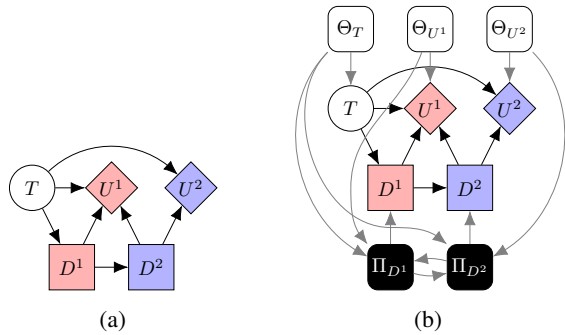

Figure 1: A causal game's (a) graph and (b) mechanised graph for Example 1.

properties of causal games and calculate certain specifications. First, in Section 3.1, we present a sound and complete set of primitive causal interventions that enable any causal intervention (a game modification) to be decomposed into one of four operations acting on CPDs or functions acting on such distributions. Second, in Section 3.2, we prove that this generalises Hammond et al. [2023]'s notion of pre-policy and post-policy interventions, which assume that interventions are either visible to all agents (pre-policy) or no agents (post-policy), to arbitrarily complex compound interventions. In Section 4, we explore how our theoretical contributions are useful for both qualitative and quantitative specifications in causal mechanism design. The former exploits graphical properties of the causal game's mechanised graph, and the latter formalises the effect of taxation and reward schemes. Finally, in Section 5, we show how causal games can be helpful for representing 'commitment', where one agent can gain a strategic advantage over others by committing to a policy before the game begins.

## 2   BACKGROUND

This section reviews Hammond et al. [2023]'s Causal Games. We begin with an example.

**Example 1** (Spence [1973]'s Job Market Signalling Game). *A worker who is either hard-working or lazy is hoping to be hired by a firm. They can choose to pursue university education but know that they will then suffer from three years of studying, especially if they are lazy. The firm prefers hard workers but is using an automated hiring system that can only observe the worker's education, not their temperament.*

We use capital letters $V$ for random variables, lowercase letters $v$ for their instantiations, and bold letters $\boldsymbol{V}$ and $\boldsymbol{v}$ respectively for sets of variables and their instantiations. We let $dom(V)$ denote the finite domain of $V$ and let $dom(\boldsymbol{V}) \coloneqq \times_{V \in \boldsymbol{V}} dom(V)$. $\mathbf{Pa}_V$ denotes the parents of variable $V$ in a graphical representation and $\mathbf{pa}_V$ the instantiation of $\mathbf{Pa}_V$. We also define $\mathbf{Ch}_V$, $\mathbf{Anc}_V$, $\mathbf{Desc}_V$, and $\mathbf{Fa}_V \coloneqq \mathbf{Pa}_V \cup \{V\}$ as the children, ancestors, descendants,

and family of $V$, respectively. As with $\mathbf{pa}_V$, their instantiations are written in lowercase. We use superscripts to indicate an agent $i \in N = \{1, \ldots, n\}$ and subscripts to index the elements of a set; for example, the decision variables belonging to agent $i$ are denoted $\boldsymbol{D}^i = \{D^i_1, \ldots, D^i_m\}$.

### 2.1   CAUSAL GAMES

Causal games (CGs) are causal multi-agent influence diagrams [Koller and Milch, 2003, Hammond et al., 2023]. Influence diagrams were initially devised to model single-agent decision problems graphically [Howard and Matheson, 2005, Miller III et al., 1976]. They are defined similarly to a Bayesian network (BN) but with additional utility variables and parameter-less decision variables. A *causal Bayesian network* (CBN) is a BN with edges that faithfully represent causal relationships [Pearl, 2009]. So, a CG is a game-theoretic CBN, where agents select a conditional distribution over actions at their decision variables to maximise the expected cumulative value of their utility variables. The simplest causal intervention $\mathrm{do}(\boldsymbol{Y} = \boldsymbol{y})$ in a CBN or CG fixes the values of variables $\boldsymbol{Y}$ to some $\boldsymbol{y}$; we denote the resulting joint distribution by $\mathrm{Pr}_{\boldsymbol{y}}(\boldsymbol{V})$.

**Definition 1.** *A **causal game (CG)** is a structure $\mathcal{M} = (\mathcal{G}, \boldsymbol{\theta})$ where $\mathcal{G} = (N, \boldsymbol{V}, \mathcal{E})$ specifies a set of agents $N = \{1, \ldots, n\}$ and a directed acyclic graph (DAG) $(\boldsymbol{V}, \mathcal{E})$ where $\boldsymbol{V}$ is partitioned into chance variables $\boldsymbol{X}$, decision variables $\boldsymbol{D} = \bigcup_{i \in N} \boldsymbol{D}^i$, and utility variables $\boldsymbol{U} = \bigcup_{i \in N} \boldsymbol{U}^i$. The parameters $\boldsymbol{\theta} = \{\theta_V\}_{V \in \boldsymbol{V} \setminus \boldsymbol{D}}$ define the CPDs $\mathrm{Pr}(V \mid \boldsymbol{Pa}_V; \theta_V)$ for each non-decision variable such that for* any *parameterisation of the decision variable CPDs, the induced model with joint distribution $\mathrm{Pr}^{\boldsymbol{\pi}}(\boldsymbol{v})$ is a causal Bayesian network, i.e., $\mathcal{G}$ is Markov compatible with $\mathrm{Pr}^{\boldsymbol{\pi}}_{\boldsymbol{y}}$ for every $\boldsymbol{Y} \subseteq \boldsymbol{V}$ and $\boldsymbol{y} \in dom(\boldsymbol{Y})$, and that:*

$$\mathrm{Pr}^{\boldsymbol{\pi}}_{\boldsymbol{y}}(v \mid \boldsymbol{pa}_V) = \begin{cases} 1 & V \in \boldsymbol{Y}, v \text{ is consistent with } \boldsymbol{y}, \\ \mathrm{Pr}^{\boldsymbol{\pi}}(v \mid \boldsymbol{pa}_V) & V \notin \boldsymbol{Y}, \boldsymbol{pa}_V \text{ is consistent with } \boldsymbol{y}. \end{cases}$$

Figure 1a depicts a causal game for Example 1. White circles represent chance variables, e.g., the worker's temperament ($T$) with probabilities $p$ for hard-working and $1 - p$ for lazy. Decision and utility variables are squares and diamonds, respectively. The worker's decision ($D^1$: attend university or not) and utility ($U^1$) are shown in red, while the firm's decision ($D^2$: offer a job or not) and utility ($U^2$) are in blue. Missing edges, like $T \to D^2$, indicate an agent's lack of information. The worker receives utility 5 for a job offer but incurs a 1 or 2 cost for attending university (depending on their temperament). The firm gains 3 for hiring a hard worker but suffers a cost of 2 if they hire a lazy worker or an opportunity cost of 1 if they reject a hard worker. Parameters $\boldsymbol{\theta}$ define conditional distributions for $T$, $U^1$, and $U^2$.

Given a causal game $\mathcal{M} = (\mathcal{G}, \boldsymbol{\theta})$, a **decision rule** $\pi_D$ for $D \in \boldsymbol{D}$ is a CPD $\pi_D(D \mid \mathbf{Pa}_D)$ and a **partial policy profile** $\pi_{\boldsymbol{D}'}$ is a set of decision rules $\pi_D$ for each $D \in$

$D' \subseteq D$, where we write $\pi_{-D'}$ for the set of decision rules for each $D \in D \setminus D'$. A **policy** $\pi^i$ refers to $\pi_{D^i}$, and a (full) **policy profile** $\pi = (\pi^1, \dots, \pi^n)$ is a tuple of policies, where $\pi^{-i} := (\pi^1, \dots, \pi^{i-1}, \pi^{i+1}, \dots, \pi^n)$. A decision rule is **pure** if $\pi_D(d \mid \mathbf{pa}_D) \in \{0, 1\}$ and **fully stochastic** if $\pi_D(d \mid \mathbf{pa}_D) > 0$ for all $d \in dom(D)$ and each **decision context** $\mathbf{pa}_D \in dom(\mathbf{Pa}_D)$; this holds for a policy (profile) if it holds for all decision rules in the policy (profile).

By combining $\pi$ with the partial distribution $\Pr$ over the chance and utility variables, we obtain a joint distribution $\Pr^{\pi}(\boldsymbol{x}, \boldsymbol{d}, \boldsymbol{u}) := \prod_{V \in \boldsymbol{V} \setminus \boldsymbol{D}} \Pr(v \mid \mathbf{pa}_V) \cdot \prod_{D \in \boldsymbol{D}} \pi_D(d \mid \mathbf{pa}_D)$ over all the variables in $\mathcal{M}$; inducing a BN. The **expected utility** for Agent $i$ given a policy profile $\pi$ is defined as the expected sum of their utility variables in this BN, that is $\mathbb{E}_{\pi}[\boldsymbol{U}^i] = \sum_{U \in \boldsymbol{U}^i} \sum_{u \in dom(U)} \Pr^{\pi}(U = u) \cdot u$. A policy $\pi^i$ is a **best response** to profile $\pi^{-i}$ if $\mathbb{E}_{(\pi^i, \pi^{-i})}[\boldsymbol{U}^i] \geq \mathbb{E}_{(\tilde{\pi}^i, \pi^{-i})}[\boldsymbol{U}^i]$ for all $\tilde{\pi}^i \in \boldsymbol{\Pi}^i$. A **Nash equilibrium** (NE) is a policy profile where each agent plays a best response. A causal game is solved by finding a policy profile that satisfies a solution concept, usually an NE.

Causal games offer several explainability and complexity advantages over extensive form games Koller and Milch [2003]. One key advantage is that probabilistic dependencies between chance and strategic variables can be exploited using the *d-separation* graphical criterion Pearl [1988].

**Definition 2.** *A **path**, $p$, in a DAG[1] $\mathcal{G} = (\boldsymbol{V}, \mathscr{E})$ is a sequence of adjacent variables in $\boldsymbol{V}$. A path $p$ is said to be **d-separated** by a set of variables $\boldsymbol{Y}$ if and only if:*

- *$p$ contains a chain $X \to W \to Z$ or $X \leftarrow W \leftarrow Z$, or a fork $X \leftarrow W \to Z$, and $W \in \boldsymbol{Y}$.*
- *$p$ contains a collider $X \to W \leftarrow Z$ and $(\{W\} \cup \boldsymbol{Desc}_W) \cap \boldsymbol{Y} = \varnothing$.*

*A set $\boldsymbol{Y}$ d-separates $\boldsymbol{X}$ from $\boldsymbol{Z}$ ($\boldsymbol{X} \perp_{\mathcal{G}} \boldsymbol{Z} \mid \boldsymbol{Y}$), if $\boldsymbol{Y}$ d-separates every path in $\mathcal{G}$ from a variable in $\boldsymbol{X}$ to a variable in $\boldsymbol{Z}$. Sets of variables that are not d-separated are said to be **d-connected**, denoted $\boldsymbol{X} \not\perp_{\mathcal{G}} \boldsymbol{Z} \mid \boldsymbol{Y}$.*

If $\boldsymbol{X} \perp_{\mathcal{G}} \boldsymbol{Z} \mid \boldsymbol{Y}$ in $\mathcal{G}$, then $\boldsymbol{X}$ and $\boldsymbol{Z}$ are probabilistically independent conditional on $\boldsymbol{Y}$ in the sense that $\Pr(\boldsymbol{x} \mid \boldsymbol{y}, \boldsymbol{z}) = \Pr(\boldsymbol{x} \mid \boldsymbol{y})$, in every distribution $\Pr$ that is Markov compatible with $\mathcal{G}$ and for which $\Pr(\boldsymbol{y}, \boldsymbol{z}) > 0$. Conversely, if $\boldsymbol{X} \not\perp_{\mathcal{G}} \boldsymbol{Z} \mid \boldsymbol{Y}$, then $\boldsymbol{X}$ and $\boldsymbol{Z}$ are dependent conditional on $\boldsymbol{Y}$ in at least one distribution Markov compatible with $\mathcal{G}$. For example, there are several paths from $U^2$ to $U^1$ in Figure 1a: direct forks through $T$ or $D^2$, a fork through $T$ and then a forward chain through $D^1$, or a backward chain through $D^2$ and then a fork through $D^1$. If $\boldsymbol{Y} = \varnothing$, then $U^2$ is d-connected to $U^1$ ($U^2 \not\perp_{\mathcal{G}} U^1 \mid \varnothing$), but if $\boldsymbol{Y} = \{T, D^2\}$ then all of the paths have been blocked by conditioning on $\boldsymbol{Y}$ and so $U^2 \perp_{\mathcal{G}} U^1 \mid \boldsymbol{Y}$.

---

[1]We use that d-separation remains a valid test for conditional independence in cyclic graphs Pearl and Dechter [1996].

A causal game's regular graph $\mathcal{G}$ captures the dependencies between **object-level** variables in the environment, but its **mechanised graph** $m\mathcal{G}$ is an enhanced representation revealing the strategically relevant dependencies between agents' decision rules and the parameterisation of the game [Hammond et al., 2023]. Collectively, decision rules and CPDs are known as the mechanisms $\mathbf{M}$ of the decision, and chance/utility variables, respectively. Each object-level variable $V \in \boldsymbol{V}$ has a mechanism parent $\mathsf{M}_V$ representing the distribution governing $V$. More specifically, each decision $D$ has a new decision rule parent $\Pi_D = \mathsf{M}_D$ and each non-decision $V$ has a new parameter parent $\Theta_V = \mathsf{M}_V$, whose values parameterise the CPDs. The *independent mechanised graph* is the result (it has no inter-mechanism edges).

However, agents select a decision rule $\pi_D$ (i.e., the value of a decision rule variable $\Pi_D$) based on both the parameterisation of the game (i.e., the values of the parameter variables) and the selection of the other decision rules $\pi_{-D}$ – so these dependencies are captured by the edges from other mechanisms into decision rule nodes. These reflect some *rationality assumptions*, captured by a set of *rationality relations* $\mathcal{R} = \{r_D\}_{D \in \boldsymbol{D}}$ that represent how the agents choose their decision rules. Each decision rule $\Pi_D$ is governed by a *serial relation* $r_D \subseteq dom(\mathbf{Pa}_{\Pi_D}) \times dom(\Pi_D)$, which accounts for the fact that an agent may not deterministically choose a single decision rule $\pi_D$ in response to some $\mathbf{pa}_{\Pi_D}$. If all of the rationality relations $\mathcal{R}$ are satisfied by $\pi$, then $\pi$ is an $\mathcal{R}$-rational outcome of the game. We often assume that the agents are playing best responses $\mathcal{R} = \mathcal{R}^{\text{BR}}$, so the $\mathcal{R}^{\text{BR}}$-rational outcomes are simply the NE of the game.

Finally, a graphical criterion $\mathcal{R}$-reachability (based on d-separation) determines which of these edges are necessary in the mechanised graph, e.g., $\mathsf{M}_V \to \Pi_D$ exists if and only if the choice of best response decision rule $\Pi_D$ depends on the CPD at $\mathsf{M}_V$ ($\mathsf{M}_V$ is $\mathcal{R}^{\text{BR}}$-relevant to $\Pi_D$). The mechanised graph for Example 1 (in Figure 1b) shows that $\Theta_T, \Theta_{U^1}$, and $\Pi_{D^2}$ are all $\mathcal{R}^{\text{BR}}$-relevant to $\Pi_{D^1}$ whereas $\Theta_T, \Theta_{U^2}$, and $\Pi_{D^1}$ are $\mathcal{R}^{\text{BR}}$-relevant to $\Pi_{D^2}$. In contrast to a causal game's regular DAG, there may exist cycles between mechanisms (see [Hammond et al., 2023] for more details).

So, the mechanised graph $m\mathcal{G}$ takes the original graph $\mathcal{G}$ and, for each variable $V \in \boldsymbol{V}$, adds mechanism parent node $\mathsf{M}_V$ and edge $\mathsf{M}_V \to V$ as well as edges $\mathsf{M}_V \to \Pi_D$ for each decision rule $\Pi_D$ where $\mathsf{M}_V$ is $\mathcal{R}^{\text{BR}}$-relevant to $\Pi_D$.

## 3 CHARACTERISING INTERVENTIONS

Causal games admit queries on level two of Pearl [2009]'s *Causal Hierarchy*. Importantly, in game-theoretic settings, we only assume that an $\mathcal{R}$-rational outcome of the game (e.g., an NE) is chosen rather than some unique policy profile $\pi$. We therefore evaluate queries with respect to a set of policy profiles, e.g., 'if $D^1 = g$, is it the case that for

all NE...'. When an intervention takes place is important. Hammond et al. [2023] previously introduced a distinction between *pre-policy* queries, where the intervention occurs before the policy profile is selected, and *post-policy* queries, where the intervention occurs after. We extend this to accommodate arbitrary queries where each agent makes decisions based on the subset of interventions visible to them.

## 3.1 PRIMITIVE INTERVENTIONS

Given a causal game $\mathcal{M}$ with mechanised graph $\mathsf{m}\mathcal{G}$ and rationality relations $\mathcal{R}$, an intervention $\mathcal{I}$ is a function that maps a set of joint probability distributions $\{\mathrm{Pr}^{\boldsymbol{\pi}}(\boldsymbol{v})\}_{\boldsymbol{\pi} \in \mathcal{R}}$ to a new set $\{\mathrm{Pr}^{\boldsymbol{\pi}}(\boldsymbol{v}_{\mathcal{I}})\}_{\boldsymbol{\pi} \in \mathcal{R}^*}$ where $\mathcal{R}^*$ are the rationality relations of the intervened game with graph $\mathsf{m}\mathcal{G}_{\mathcal{I}}$ and $\mathrm{Pr}^{\boldsymbol{\pi}}(\boldsymbol{v}_{\mathcal{I}})$ is the joint probability distribution represented by the CBN induced by $\mathsf{m}\mathcal{G}_{\mathcal{I}}$ when parameterised over policy profile $\boldsymbol{\pi}$. We define four primitive types of intervention.

**(1) Fixing an object-level variable:** Intervening on variable $X$ replaces $\mathrm{Pr}^{\boldsymbol{\pi}}(x \mid \mathbf{pa}_X)$ with a new CPD $\mathrm{Pr}^{\mathcal{I}}(x \mid \mathbf{pa}_X^*)$. Graphically, when $\mathbf{pa}_X^* \neq \mathbf{pa}_X$, the incoming edges to $X$ are changed such that $V \rightarrow X$ exists if and only if $V \in \mathbf{pa}_X^*$. The induced distribution is:

$$\mathrm{Pr}^{\boldsymbol{\pi}}(\boldsymbol{v}_{\mathcal{I}}) = \mathrm{Pr}^{\mathcal{I}}(x \mid \mathbf{pa}_X^*) \cdot \prod_{V \in \boldsymbol{V} \setminus \{X\}} \mathrm{Pr}^{\boldsymbol{\pi}}(v \mid \mathbf{pa}_V)$$

A *hard object-level intervention* assigns $\mathrm{Pr}^{\mathcal{I}} = \delta(X, g)$. In Pearl [2009]'s do-calculus, this is written $\mathrm{do}(X = g)$. Any other form of object-level intervention is qualified as *soft*.

**(2) Fixing a mechanism variable:** A *hard mechanism-level intervention* $\mathrm{do}(\mathsf{M}_V = \mathsf{m}_V)$ sets the distribution over each mechanism $\mathsf{M}_V$ to $\delta(\mathsf{M}_V, \mathsf{m}_V)$. Any other form of mechanism-level intervention is qualified as *soft*. A mechanism-level intervention on decision rule $\Pi_D$ replaces $r_D : dom(\mathbf{Pa}_{\Pi_D}) \rightarrow dom(\Pi_D)$ with a new rationality relation $r_D^{\mathcal{I}} : dom(\mathbf{Pa}_{\Pi_D}^*) \rightarrow dom(\Pi_D)$. Graphically, when $\mathbf{Pa}_{\Pi_D}^* \neq \mathbf{Pa}_{\Pi_D}$, the incoming edges to variable $\Pi_D$ are changed such that $V \rightarrow \Pi_D$ exists if and only if $V \in \mathbf{Pa}_{\Pi_D}^*$. For a parameter variable $\Theta_V$ of $V \in \boldsymbol{V} \setminus \boldsymbol{D}$, an intervention assigns a new distribution from the set of all CPDs over set $V$ given the values of its parents, set $Pa_V$. Note that parameter variables don't have parent mechanism variables as inputs to the choice of distribution.

**(3) Adding a new object-level variable:** Adding a new object-level variable $Y$ introduces a new CPD $\mathrm{Pr}^{\mathcal{I}}(y \mid \mathbf{pa}_Y)$ to the joint distribution factorisation. Graphically, this adds a new node $Y$ to $\mathcal{G}$ and adds edges $X \rightarrow Y$ for all $X \in \mathbf{Pa}_Y$ and $Y \rightarrow Z$ for all $Z \in \mathbf{Ch}_Y$. The induced distribution is

$$\mathrm{Pr}^{\boldsymbol{\pi}}((\boldsymbol{v} \cup Y)_{\mathcal{I}}) = \mathrm{Pr}^{\mathcal{I}}(y \mid \mathbf{pa}_Y) \cdot \prod_{V \in \boldsymbol{V}} \mathrm{Pr}^{\boldsymbol{\pi}}(v \mid \mathbf{pa}_V^*)$$

where, for $V \in \boldsymbol{V}$, $\mathbf{Pa}_V^* = \begin{cases} \mathbf{Pa}_V \cup Y & \text{if } V \in \mathbf{Ch}_Y \\ \mathbf{Pa}_V & \text{otherwise} \end{cases}$

**(4) Removing an existing object-level variable:** Removing an existing object-level variable $Y$ removes the CPD $\mathrm{Pr}^{\boldsymbol{\pi}}(y \mid \mathbf{pa}_Y)$ from the joint distribution factorisation. Graphically, this removes the node $Y$ from $\mathcal{G}$ and removes edges $X \rightarrow Y$ for all $X \in \mathbf{Pa}_Y$ and $Y \rightarrow Z$ for all $Z \in \mathbf{Ch}_Y$. The induced distribution is

$$\mathrm{Pr}^{\boldsymbol{\pi}}((\boldsymbol{v} \setminus \{Y\})_{\mathcal{I}}) = \prod_{V \in \boldsymbol{V} \setminus \{Y\}} \mathrm{Pr}^{\boldsymbol{\pi}}(v \mid \mathbf{pa}_V^*)$$

where, for $V \in \boldsymbol{V}$, $\mathbf{Pa}_V^* = \begin{cases} \mathbf{Pa}_V \setminus \{Y\} & \text{if } V \in \mathbf{Ch}_Y \\ \mathbf{Pa}_V & \text{otherwise} \end{cases}$

**Remark 1.** *After any intervention of type 1, 3, or 4, $\mathsf{m}\mathcal{G}$ must be updated to reflect any changes in $\mathcal{R}$-reachability between mechanisms. Note that a type 1 intervention can be considered a type 4 intervention followed by a type 3 intervention, but we include it as a primitive for convenience.*

**Theorem 1.** *Primitive interventions are a sound and complete formulation of causal interventions.*

*Proof sketch.* Soundness comes because each primitive intervention corresponds with a function between a set of probability distributions induced by $\mathcal{R}$-rational outcomes to a new set of probability distributions induced by (a possibly different) set of $\mathcal{R}$-rational outcomes. This makes it a valid causal intervention. Completeness is shown by proving any valid intervention can be decomposed into an equivalent set of primitive interventions. We relegate the full proof to Appendix A. □

There are a number of other interesting intervention types that can be constructed by composing these primitives.

**Unfixing an object-level variable:** For every type 1 intervention $\mathcal{I}$ which fixes variable $X$, there is a type 1 inverse intervention $\mathcal{I}'$ which unfixes it. It restores the intervened CPD to be based on the original policy profile $\boldsymbol{\pi}$ and parents $\mathbf{Pa}_X$, rather than $\mathcal{I}$ and $\mathbf{Pa}_X^*$.

**Unfixing a mechanism variable:** Similarly, for every type 2 intervention $\mathcal{I}$ which fixes a variable $\Pi_D$, there exists a type 2 inverse intervention $\mathcal{I}'$ which unfixes it. This restores the rationality relation associated with $\Pi_D$ to its default $r_D$, rather than $r_D^{\mathcal{I}}$. It also makes the mechanism conditionally dependent on the original parents $\mathbf{Pa}_{\Pi_D}$ rather than $\mathbf{Pa}_{\Pi_D}^*$.

**Adding an object-level dependency:** Adding a dependency, e.g., $\mathrm{add}(X \rightarrow Y)$, is equivalent to a type 1 intervention where $\mathrm{Pr}^{\mathcal{I}}(y \mid \mathbf{pa}_Y) = \mathrm{Pr}^{\boldsymbol{\pi}}(y \mid \mathbf{pa}_Y \cup \{X\})$.

**Removing an object-level dependency:** Removing a dependency, e.g., $\mathrm{del}(X \rightarrow Y)$, is equivalent to a type 1 intervention where $\mathrm{Pr}^{\mathcal{I}}(y \mid \mathbf{pa}_Y) = \mathrm{Pr}^{\boldsymbol{\pi}}(y \mid \mathbf{pa}_Y \setminus \{X\})$.

## 3.2 INTERVENTIONAL QUERIES

An interventional query concerns the outcome of a game after a set of causal interventions $\mathcal{I}$, where each agent is privy to the state of the game after a subset of these interventions has been performed. We say that an intervention is *visible* to an agent if the agent has an opportunity to adapt their policy to that intervention. Consider Example 1. Unbeknownst to the firm, the worker may have an alternative job offer which changes her best-response policy. Simultaneously, the firm may have new hiring quotas, which change their payoffs and, therefore, their best response, but which are not disclosed to the worker. These two external interventions can be expressed in a unified analysis using our framework.

First, we introduce some new notation. $\mathcal{P}$ denotes a set of primitive interventions. $\mathcal{I}^i \subseteq \mathcal{I}$ denotes the set of interventions visible to agent $i$. $\mathcal{I}(\mathcal{M})$ denotes the state of the causal game after applying interventions $\mathcal{I}$ in any order. The $\circ$ operator denotes ordered composition where $(\mathcal{I}_1 \circ \mathcal{I}_0)(\mathcal{M})$ is the state of the game after applying $\mathcal{I}_0$ then $\mathcal{I}_1$. As shorthand, $\mathcal{I}_0 \circ \mathcal{I}_1$ means $\{\mathcal{I}_0\} \circ \{\mathcal{I}_1\}$.

**Remark 2.** *The order in which interventions are applied is important because interventions are not commutative. Consider, for example, two hard object-level interventions on the same variable but to different CPDs, $\delta(X, a)$ and $\delta(X, b)$. Then clearly $(\mathrm{do}(X = a) \circ \mathrm{do}(X = b))(\mathcal{M}) \not\equiv (\mathrm{do}(X = b) \circ \mathrm{do}(X = a))(\mathcal{M})$.*

The $\mathcal{R}$-rational outcomes of the game after each agent $i$ has an opportunity to adapt to her visible interventions $\mathcal{I}^i$, is denoted $\mathcal{R}(\mathcal{M}_I)$. $\Theta_I$ denotes the parameterisation of non-decision mechanisms after interventions $\mathcal{I}$. Using this, we define an interventional query which Theorem 2 proves can always be decomposed into primitive intervention sets.

**Definition 3** (Interventional Query). *Given CG $\mathcal{M}$, rationality relations $\mathcal{R}$, and set of visible interventions for each agent $\mathcal{I}^1, \ldots, \mathcal{I}^N$, an interventional query $\phi(\pi)$ is a first-order logical formula that acts on the joint probability distribution $\Pr^{\pi}$ induced by $\mathcal{R}$-rational outcome $\pi \in \mathcal{R}(\mathcal{M}_{\mathcal{I}})$ and parameterisation $\Theta_{\mathcal{I}}$ where $\mathcal{I} = \mathcal{I}^1 \cup \ldots \cup \mathcal{I}^N$.*

**Theorem 2** (Decomposition of Intervention Sets). *For any set of interventions $\mathcal{I}$, where $\mathcal{I}^i \subseteq \mathcal{I}$ is the subset of interventions visible to agent $i$, there are primitive intervention sets $\mathcal{P}_0, \ldots, \mathcal{P}_m$, such that*

$$\forall i \, \exists j \in \{0, \ldots, m\} : \mathcal{I}^i(\mathcal{M}) = (\mathcal{P}_j \circ \mathcal{P}_{j-1} \ldots \circ \mathcal{P}_0)(\mathcal{M}) \tag{1}$$

That is to say, for any set of interventions $\mathcal{I}$, where the visible set of each agent is an *arbitrary* subset, $\mathcal{I}^i \subseteq \mathcal{I}$, we can construct an *ordered* list of primitive interventions such that, after the first $j$ sets of primitive interventions, the state of the game is the exact state visible to Agent $i$ when choosing her policy. We prove Theorem 2 in Appendix A.

Taking this decomposition, we uniquely partition the agents into sets $A_0, \ldots, A_m$ according to the state of the game visible to them. The `decompose` function maps $\mathcal{I}$ to sets $\mathcal{P}_0, \ldots, \mathcal{P}_m$ satisfying Theorem 2 and the corresponding partition of the agents $A_0, \ldots, A_m$. Then, Algorithm 1 solves the interventional query by iteratively calculating the $\mathcal{R}$-rational outcomes (e.g., NEs if $\mathcal{R} = \mathcal{R}^{\mathrm{BR}}$), fixing the policies of agents who cannot observe future interventions, and applying interventions. This subsumes Hammond et al. [2023]'s pre-policy and post-policy interventional queries. The computational complexity of Algorithm 1 is (in general) intractable, but as is almost any inference problem in Bayesian networks Kwisthout [2009]. Fox et al. [2023] discuss how algorithms such as this one will only be practical in settings with bounded tree-width graphs, number of agents, and action sets. We leave improving the efficiency of this algorithm to future work.

Whenever the rationality assumptions have a solution existence guarantee (e.g., if $\mathcal{R} = \mathcal{R}^{\mathrm{BR}}$, there is always at least one NE of the game), then Algorithm 1 successfully terminates. There are two special cases:

1. If $\mathcal{P}_0 \cup \ldots \cup \mathcal{P}_j = \emptyset$, the agent is not privy to any interventions and the interventional query is fully *post-policy* with respect to Agent $i$.

2. If $\mathcal{P}_0 \cup \ldots \cup \mathcal{P}_j = \mathcal{I}$, the agent is privy to all interventions and the interventional query is fully *pre-policy* with respect to Agent $i$.

---

**Algorithm 1** Calculate the result of an interventional query

1: **Input:** A causal game $\mathcal{M}$ with rationality relation $\mathcal{R}$, interventions $\mathcal{I} = \mathcal{I}^1 \cup \ldots \cup \mathcal{I}^N$, and query $\phi(\pi)$.
2: $(\mathcal{P}_0, \ldots, \mathcal{P}_m), (A_0, \ldots, A_m) \leftarrow \mathrm{DECOMPOSE}(\mathcal{I})$
3: $A' \leftarrow \emptyset$
4: **for** $j = 0, \ldots, m$ **do**
5: $\quad A \leftarrow (A_j \cup \ldots \cup A_N) \setminus A'$
6: $\quad \hat{\pi} \leftarrow$ uniformly sample an $\mathcal{R}$-rational outcome
7: $\quad$ **for** $i \in A_j$ **do**
8: $\quad\quad \mathrm{do}(\Pi_{D^i} = \hat{\pi}_{D^i})$
9: $\quad$ **for** $\mathcal{P} \in \mathcal{P}_j$ **do**
10: $\quad\quad \mathcal{P}(\mathcal{M})$
11: $\quad\quad$ **if** $\mathcal{P}$ acts on $V \in \boldsymbol{D}^i \cup \boldsymbol{\Pi}^i$ **then**
12: $\quad\quad\quad A' \leftarrow A' \cup \{i\}$
13: $\Pr^{\pi}(\boldsymbol{v}) \leftarrow \prod_{V \in \boldsymbol{V}} \Pr^{\pi}(v \mid \mathbf{pa}_V)$
14: **return** $\phi(\pi)$

---

**Mechanism-level side effects:** Object-level interventions can have unintuitive mechanism-level side effects. A side effect is a modification to the inter-mechanism edges in $\mathrm{m}\mathcal{G}$ and $\not\rightarrow$ denotes an edge removal. Proposition 1 formalises the side effects of an object-level intervention.

**Definition 4** (Reachability Path). *Let $D \in \boldsymbol{D}^i$. We write $\mathcal{R}(\mathrm{M}_V \rightarrow \Pi_D)$ to denote the set of paths that make $\mathrm{M}_V$*

$\mathcal{R}$-relevant to $\Pi_D$. A reachability path is any path $p \in \mathcal{R}(\mathsf{M}_V \to \Pi_D)$. *That is, a non-repeating sequence of nodes $V_0, ..., V_j \in \mathsf{m}_\perp \boldsymbol{V}$ of the independent mechanised graph $\mathsf{m}_\perp \mathcal{G}$ s.t $V_0 = \mathsf{M}_V$, and $V_0$ is $\mathcal{R}$-relevant to $\Pi_D$.*

**Proposition 1** (Object-level intervention side effects). *An object-level intervention $\mathrm{Pr}^{\mathcal{I}}(x \mid \boldsymbol{pa}_X^*)$ has side effect $\mathsf{M}_V \not\to \Pi_D$ if, $\forall$ reachability paths $p \in \mathcal{R}(\mathsf{M}_V \to \Pi_D)$ we have $\exists W \in \boldsymbol{V} s.t. (W \notin \boldsymbol{Pa}_X^*)$ and $((W \to X) \in p)$*

That is to say, an intervention on $X$ which severs at least one edge critical to each reachability path between $\mathsf{M}_V$ and $\Pi_D$ through $X$, will delete the corresponding edge between those mechanisms in $\mathsf{m}\mathcal{G}$. Similarly, if an intervention creates at least one new reachability path, it will result in the addition of a new inter-mechanism edge.

**Minimum intervention sets:** Using these observations, we formalise the minimum set of interventions required to break a causal mechanism dependency $\mathsf{M}_V \to \Pi_D$. Since we are only interested in interventions that do not directly modify the target policy $\Pi_D$, and we recall that reachability paths are calculated on the independent mechanised graph $\mathsf{m}_\perp \mathcal{G}$ which contains no edges of the form $\mathsf{M}_V \to \Pi_D$, we can restrict our attention to object-level interventions. Then, the minimum intervention set is the minimum hitting set across all reachability paths, of the variables with incoming edges that would break the dependency if removed.

**Definition 5** (Minimum intervention set). *The minimum set of objects to intervene on in order to break causal mechanism dependency $\mathsf{M}_V \to \Pi_D$ is $X$ s.t. :*

$$X \cap S_i \neq \emptyset \text{ for all } S_i \text{ where}$$
$$S_i = \{V \in \boldsymbol{V} \mid \exists W \in \boldsymbol{V}.(W \to V) \in \mathcal{R}(\mathsf{M}_V \to \Pi_D)\}$$

This metric measures how *robust* a causal mechanism dependency is to external interventions. The size of this set is the minimum number of object-level interventions required to ensure that, under every parameterization of the game, there is no incentive for a target policy $\Pi_D$ to depend on the mechanism variable $\mathsf{M}_V$.

# 4 CAUSAL MECHANISM DESIGN

Mechanism design [Börgers et al., 2015] aims to modify a game to satisfy a desired social outcome or agent behaviour. Current approaches [Peysakhovich et al., 2019, Paccagnan et al., 2022] establish error bounds on expected outcomes for particular families of games when an intervention is conducted. This section explores how our framework enables a systematic approach to causal mechanism design.

## 4.1 QUALITATIVE SPECIFICATIONS

Qualitative specifications are concerned with properties of the DAG $\mathcal{G}$. Consider the mechanised graph of Example 1

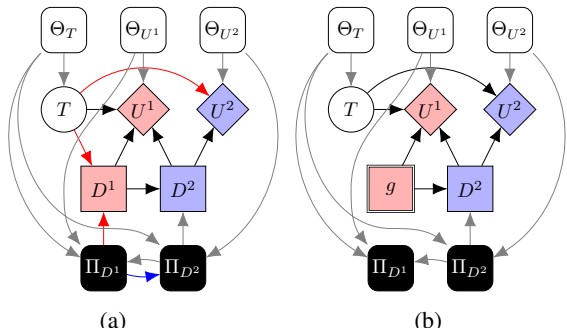

Figure 2: The *Job Market* game with an intervention $\mathrm{do}(D^1 = g)$ satisfying qualitative specification $\Pi_{D^1} \not\to \Pi_{D^2}$. Mechanism-level dependencies are coloured grey. In (a), the blue edge indicates the $\mathcal{R}^{\mathrm{BR}}$-relevance of $\Pi_{D^1}$ to $\Pi_{D^2}$ and red edges indicate an active reachability path. In (b), the intervention $\mathrm{do}(D^1 = g)$ breaks all the reachability paths which made $\Pi_{D^1}$ relevant to $\Pi_{D^2}$.

shown in Figure 2a. The cyclic structure between nodes $\Pi_{D^1}$ and $\Pi_{D^2}$ means the optimal policy for each agent depends on the other agent's policy.

A specification may require a decision rule to be *independent* of a particular mechanism. For example, we may want the firm's hiring policy to be independent of the worker's policy when deciding to go to university. That is, we wish to break the causal dependency $\Pi_{D^1} \to \Pi_{D^2}$. When this edge does not exist, it means that the firm's optimal policy does not depend on the worker's policy for *any* parameterisation of the game. There are two ways to satisfy this specification.

1. Intervene on the target policy $\Pi_{D^2}$ with $r_{D^2}^{\mathcal{I}} : dom(\mathbf{Pa}_{\Pi_{D^2}}^*) \to dom(\Pi_{D^2})$ such that $\Pi_{D^1} \notin \mathbf{Pa}_{\Pi_{D^2}}^*$, e.g., the hard intervention $\mathrm{do}(\Pi_{D^2} = \delta(D^2, \neg j))$ which forces the firm to reject every candidate.

2. Perform an object-level intervention to appropriately change the reachability structure of the graph. There are two paths that make $\Pi_{D^1}$ $\mathcal{R}^{\mathrm{BR}}$-relevant to $\Pi_{D^2}$. The first is $\Pi_{D^1} \to D^1 \leftarrow T \to U^2$ when conditioned on $\{D^2, D^1\}$ since $\Pi_{D^1} \not\perp_{\mathsf{m}_\perp \mathcal{G}} U^2 \cap \mathbf{Desc}_{D^2} \mid D^2, \mathbf{Pa}_{D^2}$. The second is $\Pi_{D^1} \to D^1$ conditioned on $\emptyset$ since $\Pi_{D^1} \not\perp_{\mathsf{m}_\perp \mathcal{G}} \mathbf{Pa}_{D^2}$.

Option 1 is somewhat against the "spirit" of mechanism design, which seeks to induce certain behaviours or social outcomes without undermining an agent's ability to make their own rational choices. However, the intervention on $\Pi_{D^2}$ changes properties of the target agent's behaviour by directly intervening on their policy.

Option 2 requires both active paths to be blocked. This can be achieved through an intervention on $D^1$ of the form $\mathrm{Pr}^{\mathcal{I}}(d^1 \mid \mathbf{pa}_{D^1}^*)$ where $\mathbf{Pa}_{D^1} = \emptyset$. An example would be $\mathrm{do}(D^1 = g)$, shown in Figure 2b. The cyclic structure between $\Pi_{D^1}$ and $\Pi_{D^2}$ is broken, and the firm has no incentive

to consider the worker's policy.

**Hiding and Revealing Information**   Another qualitative specification is to hide or reveal certain information to agents. This can be done by modifying the incoming edges into a decision variable. Suppose we wish to hide the agent's decision of going to university from the firm in Example 1. Intervention $\text{del}(D^1 \rightarrow D^2)$ satisfies this but has mechanism level side-effect $\Pi_{D^1} \not\rightarrow \Pi_{D^2}$.

A more general question is: under what circumstances is it possible to hide or reveal information *without* changing the mechanism dependency structure? The mechanism dependency structure is retained if, for any pair of mechanisms with active reachability paths, at least one path is not broken, and if, for any pair of mechanisms with no active reachability paths, no new paths are introduced. We call an intervention that preserves this structure *incentive invariant*.

**Definition 6** (Incentive Invariance). *An intervention $\mathcal{I}$ is incentive invariant if $\forall\, \mathsf{M}_V \in \mathbf{M}, \forall\, \Pi_D \in \mathbf{\Pi}$, we have pre-intervention reachability paths $\mathcal{R}^{BR}(\mathsf{M}_V \rightarrow \Pi_D)$ and post-intervention reachability paths $\mathcal{R}^{BR*}(\mathsf{M}_V \rightarrow \Pi_D)$ s.t.*

$$|\mathcal{R}^{BR*}(\mathsf{M}_V \rightarrow \Pi_D)| \begin{cases} = 0, \text{ if } |\mathcal{R}^{BR}(\mathsf{M}_V \rightarrow \Pi_D)| = 0 \\ > 0, \text{ if } |\mathcal{R}^{BR}(\mathsf{M}_V \rightarrow \Pi_D)| > 0 \end{cases}$$

### 4.2   QUANTITATIVE SPECIFICATIONS

Quantitative specifications describe bounds on game outcomes. For example, they specify that the expected payoff of an agent is greater than some value, that the probability of a certain event occurring is within some range, or that some social welfare metric is maximised. There are many ways of satisfying these specifications, including the modifications to the object-level and mechanism-level dependencies discussed previously. Here, we focus on interventions that directly modify the chance or utility variables of the game or the corresponding mechanism-level parameter variables.

**Taxes and Rewards:** One way of inducing certain behaviour is to modify the payoffs for certain outcomes through taxes and rewards. The inter-mechanism edges reveal which utility variables, under some parameterisation of the game, can affect an agent's $\mathcal{R}$-rational choice of policy. For example, consider the Prisoner's Dilemma, where the prisoners are restricted to pure policies. The mechanised graph for this is the same as in Figure 3a. Suppose we are a sadistic game designer who wants to maximise the jail time of both prisoners by any means. We can do this in several ways by modifying the usual payoffs of the game (Table 1).

One way is to decrease the payoffs of the NE $(D, D)$ (the $\mathcal{R}^{BR}$-rational outcome). Typically, mutual defection leads to a total jail time of 4 years. Changing $(D, D)$ to $(-3, -3)$ yields 6 years total. In fact, we could change the payoff of $(D, D)$ to $(-5 + \varepsilon, -5 + \varepsilon)$ for arbitrary $\varepsilon > 0$ yielding

Table 1: The payoffs in the Prisoner's Dilemma.

|  |  | Agent 2 (Bob) | |
|---|---|---|---|
|  |  | Cooperate | Defect |
| Agent 1 (Alice) | Cooperate | (-1, -1) | (-5, 0) |
|  | Defect | (0, -5) | (-2, -2) |

$-10 + 2\varepsilon$ years total while retaining $(D, D)$ as the single pure policy NE. Since this intervention does not affect the best-response of either prisoner, it doesn't matter whether this intervention is implemented as a fully pre-policy, fully post-policy, or interleaved intervention. The prisoners will play the same policies and the same NE will be reached.

Another way is by *taxing* the existing rational outcome. In fact, by introducing a partially visible intervention, we can also *reward* certain behaviours to satisfy the specification. If Alice believes that mutual cooperation will lead to both agents going free, while Bob believes they will suffer 1 year each, then $(C, D)$ becomes a new NE. This can be implemented in one of two ways.

**Example 2** (Partially Visible Rewards). *We want to influence one prisoner in the Prisoner's Dilemma to cooperate. C and D indicate the pure policies "cooperate" and "defect" respectively. Let $\mathcal{P} = \{\text{do}(\Theta_{U^1} = \theta^*_{U^1}), \text{do}(\Theta_{U^2} = \theta^*_{U^2})\}$ be the set of primitive interventions with*

$$\theta^*_{U^1}(u^1 \mid d^1, d^2) = \begin{cases} \delta(u^1, 0) & \text{if } d^1 = C \text{ and } d^2 = C \\ \theta_{U^1}(u^1 \mid d^1, d^2) & \text{otherwise} \end{cases}$$

*and similar for $\theta^*_{U^2}$. Let $\mathcal{P}'$ be the inverse. We make $\mathcal{P}$ visible to only Alice in one of two ways:*

1. *$\mathcal{P}_0 = \emptyset$, $\mathcal{P}_1 = \mathcal{P}$, $A_0 = \{Bob\}$, $A_1 = \{Alice\}$. This changes the payoffs of $(C, C)$ so both prisoners go free, but only Alice is informed of the change (the intervention is hidden from Bob).*

2. *$\mathcal{P}_0 = \mathcal{P}$, $\mathcal{P}_1 = \mathcal{P}'$, $A_0 = \{Alice\}$, $A_1 = \{Bob\}$. This informs Alice that $(C, C)$ will lead to both prisoners going free but reverses this intervention between Alice's and Bob's policy choices, so it deceives Alice into believing an intervention has taken place.*

*In either case, Alice believes there are two possible NE: $(C, C)$ and $(D, D)$, whereas Bob believes there is only one $(D, D)$. So, if Alice plays uniform distribution over her best responses $C$ and $D$, and Bob plays $\delta(D^2, D)$, the expected total jail time is $\mathbb{E}_{\hat{\pi}}[U^1 + U^2] = \frac{1}{2}(0 - 5) + \frac{1}{2}(-2 - 2) = -4.5$ Therefore, adding total reward of 2 to game outcome reduces the expected total payoff by 0.5.*

**Environment Modifications:** Another way of satisfying a quantitative specification is to modify the chance variables. In Example 1, the worker's temperament can affect both agents' policies. Suppose we want to maximise the probability of the worker getting a job and $\mathcal{R} = \mathcal{R}^{BR}$, i.e., we

want the probability of the worker getting the job under any NE of the intervened game to be at least as high as the probability of the worker getting the job under any NE of the original game. Formally, an intervention $\mathcal{I}$ satisfies this specification if

$$\min_{\hat{\pi} \in \mathcal{R}^{\mathrm{BR}}(\mathcal{M}_{\mathcal{I}})} \Pr^{\hat{\pi}}(j) \geq \max_{\pi \in \mathcal{R}^{\mathrm{BR}}(\mathcal{M})} \Pr^{\pi}(j)$$

One way to do this is to change the location of the game to *EffortVille* where everyone is hard-working. This corresponds with a mechanism-level $\mathrm{do}(\Theta_T = \delta(T, h))$ or object-level $\mathrm{do}(T = h)$ intervention. In this case, the CG has three pure policy $\mathcal{R}^{\mathrm{BR}}$-rational outcomes (NE).

1. The worker always chooses $g$. The hiring system always chooses $j$. So $\mathbb{E}_{\pi}[U^1] = 5$ and $\mathbb{E}_{\pi}[U^2] = 3$

2. The worker always chooses $g$. The hiring system chooses $j$ if the worker chooses $g$. Otherwise, it chooses $\neg j$. So $\mathbb{E}_{\pi}[U^1] = 5$ and $\mathbb{E}_{\pi}[U^2] = 3$

3. The worker always chooses $\neg g$. The hiring system chooses $\neg j$ if the worker chooses $g$. Otherwise, it chooses $j$. So $\mathbb{E}_{\pi}[U^1] = 4$ and $\mathbb{E}_{\pi}[U^2] = 3$

In all these NEs, the probability of the worker getting a job is 1, so it satisfies the specification. Also, the first two NEs of the intervened game maximise utilitarian and egalitarian social welfare. The identity intervention, which does not change the game, would also have satisfied this specification because all three pure NEs of the original game also result in the worker getting a job with probability 1. However, this is not the case under NEs with stochastic policies. The original game has the following NE: If the worker is hardworking, she chooses $g$ with probability $\frac{1}{2}$. If she is lazy, she always chooses $\neg g$. If she chooses $g$, the firm always chooses $j$. If she chooses $\neg g$, the firm chooses $j$ with probability $\frac{4}{5}$. This yields a $\frac{9}{10}$ probability of the worker getting a job. On the other hand, the NEs of the intervened game are

1. The worker always chooses $g$, The hiring system chooses $j$ if the worker chose $g$, otherwise it chooses $j$ with any probability $q_1 \in [0, 1]$.

2. The worker always chooses $\neg g$, The hiring system chooses $j$ if the worker chose $\neg g$, otherwise it chooses $j$ with any probability $q_2 \in [0, \frac{4}{5}]$.

So, the worker gets a job with probability 1 either way. Therefore, the intervention of *EffortVille* satisfies the specification in the stochastic policy case, whereas the identity intervention does not. In our intervention framework, we can model *EffortVille* with primitive intervention set $\mathcal{P}_0 = \{\mathrm{do}(T = h)\}$ and $A_0 = \{1, 2\}$ since we want the intervention to be fully pre-policy, allowing both agents to adapt their policies accordingly. Note, however, that it is typically not possible for game designers to intervene on the chance variables of the game as these are usually used to represent 'moves by nature'. For example, a government may be able to intervene on utility variables by taxing or rewarding workers and firms, but it is unlikely that they can affect the underlying temperament of the workers.

# 5 COMMITMENT

Interventions on decision and decision rule variables enable us to reason about *commitment*. In some games, it is possible for the first moving agent, the *leader*, to gain a strategic advantage over others, called *followers*, by *committing* to a policy before the game begins; the leader can sometimes influence the follower's incentives by revealing private information about their policy. The simplest example is a *Stackelberg game* consisting of one leader and follower.

We use an example from [Letchford and Conitzer, 2010], which shares the same game graph as in Figure 3a with Agent 1 (2) the leader (follower) and with $dom(D^1) = \{T, B\}$ and $dom(D^2) = \{L, R\}$. The utility parameterization is shown in Table 3c. Pre-commitment, Action $T$ strictly dominates $B$ so $(T, L)$ is a unique NE and $\mathbb{E}_{\pi}[U^1] = 2$. However, by committing to the pure policy $B$, the leader incentivises the follower to play $R$ and so $\mathbb{E}_{\pi}[U^1] = 3$. Note, in this case, the result of commitment is also a Pareto improvement over the original NE (i.e., Stackelberg commitment can also improve social welfare).

Causal games naturally represent commitment with a simple causal intervention on node $\Pi_{D^1}$ to be fixed to the committed policy $\pi_1$ (shown in Figure 3b). The payoff received by the leader after commitment can be calculated through backward induction on the graph [Hammond et al., 2021].

By representing commitment as a causal intervention, we can prove whether a particular commitment can be beneficial for the leader. In the stochastic policy setting, the follower will still play a pure policy since she has no incentive to randomise after the leader's commitment; she is effectively playing a single-agent decision game. The leader's expected utility after committing to policy $\pi_1 = \frac{1}{2}T + \frac{1}{2}B$ is 3.5 (in Appendix B.1). This is greater than the expected utility of 2 in the original game's unique NE, which benefits the leader.

A partially visible commitment is represented naturally in our new framework of causal interventions. Specifically, a commitment that occurs in the primitive intervention set $\mathcal{P}_j$ can be revealed to all agents in $A_j, A_{j+1}, \ldots, A_m$. For example, if we have $\mathcal{P}_0 = \emptyset$, $\mathcal{P}_1 = \{\mathrm{do}(\Pi_1 = \delta(D^1, T))\}$, $A_0 = \{1\}$, and $A_1 = \{2\}$ then Agent 1 commits to playing 'B' and reveals it to Agent 2. Algorithm 1 reveals that Agent 2 will play $\delta(D^2, R)$ (i.e. always playing "R") and Agent 1 will receive a payoff of 3. However, if Agent 1's commitment to playing $\delta(D^1, B)$ is kept private from Agent 2, then we have $\mathcal{P}_0 = \emptyset$, $\mathcal{P}_1 = \{\mathrm{do}(\Pi_1 = \delta(D^1, B))\}$, $A_0 = \{1, 2\}$, and $A_1 = \emptyset$. Then, Agent 2 will always play 'L', in accordance with the NE of the original system,

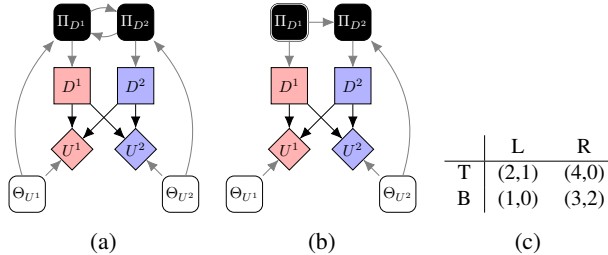

|   | L | R |
|---|---|---|
| T | (2,1) | (4,0) |
| B | (1,0) | (3,2) |

(a)      (b)      (c)

Figure 3: The Stackleberg game represented as mechanised causal graphs (a) pre-commitment, and (b) post-commitment, with payoff matrix (c).

calculated after $\mathcal{P}_0$, giving Agent 1 a payoff of 2. In Appendix B.2, we show that we can also use the intervened graph to calculate the *optimal* policy to commit to.

# 6 CONCLUSION

This work presents a sound and complete characterisation of arbitrary causal interventions in causal games. It uses this framework to evaluate and systematically modify incentive structures to satisfy qualitative and quantitative specifications, which has important applications for causal mechanism design. Solving interventional queries is computationally expensive, but we prove results and give algorithms, showing how they can be made more tractable. Finally, we focus on pedagogical examples, but demonstrating the method empirically on larger examples is an important direction for future work.

### Acknowledgements

The authors wish to thank five anonymous reviewers for their helpful comments. Fox was supported by the EPSRC Centre for Doctoral Training in Autonomous Intelligent Machines and Systems (Reference: EP/S024050/1) and Wooldridge was supported by a UKRI Turing AI World Leading Researcher Fellowship (Reference: EP/W002949/1).

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

# Characterising Interventions in Causal Games
## (Supplementary Material)

**Manuj Mishra**[1]          **James Fox**[1,2]          **Michael Wooldridge**[1]

[1]Department of Computer Science, University of Oxford, UK
[2]London Initiative for Safe AI, UK

## A   PROOFS

**Theorem 1.** *Primitive interventions are a sound and complete formulation of causal interventions.*

*Proof.* We first prove soundness. This is true using the definitions of the primitive interventions. Let $\Pr(\boldsymbol{v}_{\mathcal{I}})$ be the induced joint distribution by type 1, 3, and 4 interventions, as per the definitions, and $\Pr(\boldsymbol{v}_{\mathcal{I}}) = \Pr(\boldsymbol{v})$ for type 2 interventions. Also, let $\mathcal{R}^*$ be the induced rationality relations by type 2 interventions and $\mathcal{R}^* = \mathcal{R}$ for type 1, 3, and 4 interventions. Then, the effect of a primitive intervention P of any type is the function $\{\Pr^{\boldsymbol{\pi}}(\boldsymbol{v})\}_{\boldsymbol{\pi} \in \mathcal{R}} \mapsto \{\Pr^{\boldsymbol{\pi}}(\boldsymbol{v}_{\mathcal{I}})\}_{\boldsymbol{\pi} \in \mathcal{R}^*}$ which is a valid causal intervention.

We now turn to completeness by showing that any intervention $\mathcal{I}$ can be decomposed into a set of equivalent primitive interventions $\mathcal{P}$. That is to say, the state of the game after applying intervention $\mathcal{I}$ is equivalent to the state of the game after applying interventions $\mathcal{P}$. Suppose,

$$\mathcal{I}(\{\Pr^{\boldsymbol{\pi}}(\boldsymbol{v})\}_{\boldsymbol{\pi} \in \mathcal{R}}, \mathcal{R}) = (\{\Pr^{\boldsymbol{\pi}}(\boldsymbol{v}_{\mathcal{I}})\}_{\boldsymbol{\pi} \in \mathcal{R}^*}, \mathcal{R}^*)$$

$$\text{s.t. } \Pr^{\boldsymbol{\pi}}(\boldsymbol{v}) = \prod_{V \in \boldsymbol{V}} \Pr^{\boldsymbol{\pi}}(v \mid \mathbf{pa}_V)$$

$$\text{and } \Pr^{\boldsymbol{\pi}}(\boldsymbol{v}_{\mathcal{I}}) = \prod_{V_{\mathcal{I}} \in \boldsymbol{V}_{\mathcal{I}}} \Pr^{\boldsymbol{\pi}}(v_{\mathcal{I}} \mid \mathbf{pa}_V)$$

Then the trivial decomposition of $\mathcal{I}$ is $|\boldsymbol{V}_{\mathcal{I}}|$ type 3 interventions which multiply the joint distribution by each of $\Pr^{\boldsymbol{\pi}}(v_{\mathcal{I}} \mid \mathbf{pa}_V)$, followed by $|\boldsymbol{V}|$ type 4 interventions that divide the joint distribution by each of $\Pr^{\boldsymbol{\pi}}(v \mid \mathbf{pa}_V)$, then $|\mathcal{R}^*|$ type 2 interventions that attach the appropriate rationality relation to each mechanism variable.

Of course, more concise decompositions are possible if there is overlap between $\mathcal{R}$ and $\mathcal{R}^*$ as well as overlap between $\{\mathbf{Pa}_V\}_{V \in \boldsymbol{V}}$ and $\{\mathbf{Pa}_{V_{\mathcal{I}}}\}_{V_{\mathcal{I}} \in \boldsymbol{V}_{\mathcal{I}}}$. $\qquad\square$

**Proposition 1** (Object-level intervention side effects). *An object-level intervention $\Pr^{\mathcal{I}}(x \mid \boldsymbol{pa}_X^*)$ has side effect $\mathsf{M}_V \nrightarrow \Pi_D$ if, $\forall$ reachability paths $p \in \mathcal{R}(\mathsf{M}_V \to \Pi_D)$ we have $\exists W \in \boldsymbol{V}$ s.t.$(W \notin \boldsymbol{Pa}_X^*)$ and $((W \to X) \in p)$*

*Proof.* An intervention $\Pr^{\mathcal{I}}(x \mid \mathbf{pa}_X^*)$ has side effect $\mathsf{M}_V \nrightarrow \Pi_D$ if in the intervened graph there are no reachability paths from $\mathsf{M}_V$ to $\Pi_D$. This means at least one causal arrow is broken in each such reachability path in the original graph. An object-level intervention on $X$ breaks only the causal arrows $W \to X$ where $W \in \mathbf{Pa}_X$ but $W \notin \mathbf{Pa}_X^*$. $\qquad\square$

**Theorem 2** (Decomposition of Intervention Sets). *For any set of interventions $\boldsymbol{\mathcal{I}}$, where $\boldsymbol{\mathcal{I}}^i \subseteq \boldsymbol{\mathcal{I}}$ is the subset of interventions visible to agent $i$, there are primitive intervention sets $\boldsymbol{\mathcal{P}}_0, \dots, \boldsymbol{\mathcal{P}}_m$, such that*

$$\forall i \; \exists j \in \{0, \dots, m\} : \boldsymbol{\mathcal{I}}^i(\mathcal{M}) = (\boldsymbol{\mathcal{P}}_j \circ \boldsymbol{\mathcal{P}}_{j-1} \dots \circ \boldsymbol{\mathcal{P}}_0)(\mathcal{M}) \tag{2}$$

*Proof.* We show there is a set of primitive intervention sets that satisfy Theorem 2 for any set of interventions $\mathcal{I}^i$ by constructing an example. We use the notation $\mathcal{P}(\mathcal{I})$ to denote the primitive decomposition of $\mathcal{I}$ as shown in the proof of Theorem 1. Then, $\mathcal{P}^i = \cup_{\mathcal{I} \in \mathcal{I}^i} \mathcal{P}(\mathcal{I})$ are the primitive interventions equivalent to each agent's visible interventions. Consider an arbitrary ordering of $\mathcal{P}^i = \mathcal{P}_0^i, \ldots, \mathcal{P}_k^i$. Let $\mathcal{Q}_k^i$ denote the inverse of $\mathcal{P}_k^i$. The construction is as follows.

$$\mathcal{P}_0 = \mathcal{P}^0$$
$$\mathcal{P}_j = \mathcal{P}_k^j \circ \ldots \circ \mathcal{P}_0^j \circ \mathcal{Q}_0^{j-1} \circ \ldots \circ \mathcal{Q}_k^{j-1} \text{ for } j \in 1, \ldots, N$$
$$A_0 = \emptyset$$
$$A_j = \{j\} \text{ for } j \in 1, \ldots, N$$

where $\mathcal{P}^0$ are the fully pre-policy interventions visible to all agents. In this construction, $A_j$ are singleton sets (except $A_0$) so we have $m = N$. For all $i$, let $j = i$. Then, we make an inductive argument. The base case $\mathcal{I}^0(\mathcal{M}) = \mathcal{P}^0(\mathcal{M}) = \mathcal{P}_0(\mathcal{M})$ holds by definition of $\mathcal{P}^0$. Assuming $\mathcal{I}^{i-1}(\mathcal{M}) = (\mathcal{P}_{j-1} \circ \mathcal{P}_{j-2} \ldots \circ \mathcal{P}_0)(\mathcal{M})$, we have

$$\mathcal{I}^i(\mathcal{M}) = \mathcal{P}^i(\mathcal{M})$$
$$= (\mathcal{P}^i \circ \mathcal{Q}^{i-1} \circ \mathcal{P}^{i-1})(\mathcal{M})$$
$$= (\mathcal{P}_j \circ \mathcal{P}^{i-1})(\mathcal{M})$$
$$= (\mathcal{P}_j \circ \mathcal{I}^{i-1})(\mathcal{M})$$
$$= (\mathcal{P}_j \circ \mathcal{P}_{j-1} \ldots \circ \mathcal{P}_0)(\mathcal{M})$$

So, this assignment of primitive intervention sets satisfies Theorem 2. Algorithm 1 explicitly shows how this construction is used to calculate interventional queries. $\square$

## B COMMITMENT

### B.1 EXPECTED UTILITY AFTER COMMITMENT

We first show that the expected utility to agent 1 is 3.5 after she commits to policy $\pi_1$, picking between $T$ and $B$ with equal probability.

Let $\mathcal{I} = \{\text{do}(\Pi_1 = \pi_1)\}$ and fix $\boldsymbol{\pi} \in \boldsymbol{\Pi}$. We use $\text{Pr}^{\mathcal{I}}(X)$ as shorthand for $\text{Pr}^{\boldsymbol{\pi}}(X_{\mathcal{I}})$, and $\mathbb{E}_{\mathcal{I}}$ as shorthand for $\mathbb{E}_{\boldsymbol{\pi}}[U_{\mathcal{I}}]$. Then

$$\mathbb{E}_{\mathcal{I}}[U^1] := \sum_{u^1 \in dom(U^1)} u^1 \text{Pr}^{\mathcal{I}}(u^1) \tag{E1}$$
$$= 2 \cdot \text{Pr}^{\mathcal{I}}(D^1 = T \mid \pi_1)\text{Pr}^{\mathcal{I}}(D^2 = L \mid \Pi_{D^2})$$
$$+ 4 \cdot \text{Pr}^{\mathcal{I}}(D^1 = T \mid \pi_1)\text{Pr}^{\mathcal{I}}(D^2 = R \mid \Pi_{D^2})$$
$$+ 1 \cdot \text{Pr}^{\mathcal{I}}(D^1 = B \mid \pi_1)\text{Pr}^{\mathcal{I}}(D^2 = L \mid \Pi_{D^2})$$
$$+ 3 \cdot \text{Pr}^{\mathcal{I}}(D^1 = B \mid \pi_1)\text{Pr}^{\mathcal{I}}(D^2 = R \mid \Pi_{D^2})$$
$$\text{Pr}^{\mathcal{I}}(D^2 = L \mid \Pi_{D^2}) = \begin{cases} 1 & \text{if } 0.5 \cdot U^2(2,1) + 0.5 \cdot U^2(1,0) \\ & \quad > 0.5 \cdot U^2(4,0) + 0.5 \cdot U^2(3,2) \\ 0 & \text{otherwise} \end{cases}$$
$$= \begin{cases} 1 & \text{if } 0.5 > 1 \\ 0 & \text{otherwise} \end{cases}$$
$$= 0$$
$$\text{Similarly, } \text{Pr}(D^2 = R \mid \Pi_{D^2}) = 1$$
$$\implies \mathbb{E}_{\mathcal{I}}[U^1] = 4 \cdot 0.5 + 3 \cdot 0.5 = 3.5$$

## B.2 OPTIMAL STOCHASTIC BEHAVIOURAL POLICY

Let agent 1 have policy $\pi_1$ where she plays $T$ with probability $p$ and agent 2 have policy $\pi_2$ where she plays $L$ with probability $q$. Then the optimal policy $\hat{\pi}_1 = \hat{p}T + (1-\hat{p})B$ is given by:

$$
q = \begin{cases} 1 & \text{if } p > 2(1-p) \\ 0 & \text{otherwise} \end{cases}
$$

$$
= \begin{cases} 1 & \text{if } p > \frac{2}{3} \\ 0 & \text{otherwise} \end{cases}
$$

$$
\implies \hat{p} = \operatorname*{argmax}_p \mathbb{E}_{\mathcal{I}}[U^1] \tag{E2}
$$

$$
= \operatorname*{argmax}_p (2pq + 4p(1-q) + (1-p)q + 3(1-p)(1-q))
$$

$$
= \operatorname*{argmax}_p (p - 2q + 3)
$$

$$
= \operatorname*{argmax}_p \left( p - 2\mathbb{I}\left( p > \frac{2}{3} \right) + 3 \right)
$$

$$
= \frac{2}{3}
$$

where $\mathcal{I} = \{\mathrm{do}(\Pi_1 = \hat{\pi}_1)\}$.

So the optimal policy for agent 1 to commit to is $\hat{\pi}_1 = \frac{2}{3}T + \frac{1}{3}B$ with payoff $\mathbb{E}_{\mathcal{I}}[U^1] = \frac{2}{3} + 3 = 3.\dot{6}$. This is greater than the payoff of 2 in the NE of the original game and the payoff of 3.5 in the NE induced after a commitment to $\frac{1}{2}T + \frac{1}{2}B$ as shown in the previous section.