# OpenReview forum: "Characterising Interventions in Causal Games"
_auai.org/UAI/2024/Conference — UAI 2024 oral_

### Official Review · Reviewer_jtKN · 2024-03-06

**Q2-1 Originality-Novelty:** 3
**Q2-2 Correctness-Technical Quality:** 3
**Q2-5 Clarity Of Writing:** 3

**Q1 Summary And Contributions:**

The paper examines causal influence diagrams and explores how various types of interventions can be simulated within these models. This simulation aims to facilitate interventional queries in multi-agent causal gaming environments.

**Q2-3 Extent To Which Claims Are Supported By Evidence:**

3: Good: the main claims are supported by convincing evidence (in the form of adequate experimental evaluation, proofs, (pseudo-)code, references, assumptions).

**Q2-4 Reproducibility:**

3: Good: key resources (e.g. proofs, code, data) are available and key details (e.g. proofs, experimental setup) are sufficiently well-described for competent researchers to confidently reproduce the main results.

**Q3 Main Strengths:**

Good methodological description. Claims are clear and supported. Excellent writing and detailed descriptions. Good level of novelty in terms of investigating the usefulness of causal influence diagrams in the context of multi-agent game theory.

**Q4 Main Weakness:**

Presentation of ideas could improve to make it easier for the readers to understand the contributions of this paper. The level of novelty in terms of simulation of interventions is not made entirely clear and seems to be restricted to simply characterising different types of causal interventions in the context of multi-agent games.

**Q5 Detailed Comments To The Authors:**

This is a good paper that effectively explains the concepts of causal influence diagrams and game theory within a unified framework. However, it would benefit from a more explicit articulation of its contributions and novelty. Currently, it's ambiguous whether the simulation of interventions for optimal decision-making presented here mirrors approaches from other fields or if the paper introduces a novel framework for simulating specific types of interventions. This is the only reason that keeps me from suggesting a stronger 'accept'.

**Q9 Complying With Reviewing Instructions:**

Yes

---

> ### Author Rebuttal · Authors · 2024-04-08
>
> We have now articulated our novel contribution better in the contributions section. Existing work assumes that an intervention is either fully post-policy (entirely invisible) to all agents or fully pre-policy (entirely visible) to all agents before they decide on their decision rule at each decision point. Our most important novel contribution is to extend the theory to be able to accommodate arbitrary queries where agents choose their decision rules based on any subset of the interventions (those visible to them). This is necessary to discuss more properties of causal games (e.g. Definition 6) and then calculate certain specifications (e.g. Example 2).

---

### Official Review · Reviewer_m4gw · 2024-03-18

**Q2-1 Originality-Novelty:** 3
**Q2-2 Correctness-Technical Quality:** 3
**Q2-5 Clarity Of Writing:** 4

**Q10 Ethical Concerns:**

No, this paper studies a causal framework that accounts for game-theoretic considerations, and its emphasis is on formal modeling and theory.

**Q1 Summary And Contributions:**

The paper considers a recently introduces framework of causal games - an extension of influence diagrams to multi-agent settings. The authors provide characterization results regarding the type of interventions in such a framework. More specifically, they provide a sound and a complete set of primitive causal interactions. This extends the results from prior work that considered so called pre- and post-policy interventions. The paper additionally provides some insights about causal mechanism design.

**Q2-3 Extent To Which Claims Are Supported By Evidence:**

3: Good: the main claims are supported by convincing evidence (in the form of adequate experimental evaluation, proofs, (pseudo-)code, references, assumptions).

**Q2-4 Reproducibility:**

4: Excellent: key resources (e.g. proofs, code, data) are available and key details (e.g. proof sketches, experimental setup) are comprehensively described for competent researchers to confidently and easily reproduce the main results.

**Q3 Main Strengths:**

- The paper is clearly written and interesting to read. It provides a good overview of the background on causal games needed for understanding the main claims in the paper. That said, there are a few typos in the paper, and given that the notation is somewhat heavy, it does take some time to fully comprehend the relevant concepts. I provide minor suggestions in the comments to the authors.

- The paper introduces new types of interventions in causal games, providing a set of sound interventions that are sufficient for constructing any sound intervention. The paper also provides an algorithm for answering causal queries. These results, formalized in Theorem 1 and Theorem 2, appear to be the main contributions of this work.

- I also appreciate the part of the paper that views intervention in causal games from the perspective of mechanism design.

**Q4 Main Weakness:**

- While I found this framework interesting, the current set of results don't demonstrate the utility of the framework relative to standard game-theoretic frameworks (similarly for the mechanism design part of the paper). E.g., I was expecting to see some results related to identifiability of causal effects, which I think would be quite interesting, but perhaps also non-trivial to obtain given that one needs to account for game-theoretic considerations (e.g., multiplicity of NE).

- The paper does not discuss computational aspects. There are two points I would like to make. First, regarding rational relations and outcomes, it's not clear to me under which conditions rational outcomes exists, unless $\mathcal R = \mathcal R^{BR}$. In that regard, it's not clear why we consider $\mathcal R \ne \mathcal R^{BR}$ in this work. The second point is related to the computation complexity of Algorithm 1. It seems that Algorithm 1 solves NE multiple times ($\mathcal R = \mathcal R^{BR}$), which is a PPAD-hard problem, so it's not clear whether the algorithm is indeed practical.

- The examples provided are rather illustrative. It would be useful to include more realistic experimental testbeds and showcase the utility of the approach through experiments.

**Q5 Detailed Comments To The Authors:**

Please see my comments above. The author response to those would be welcome. E.g., it would be great if you could comment on the computational complexity of performing interventional queries, and the tractability of Algorithm 1. Some minor remarks and suggestions:
- It would be also valuable to compare this work with the line of work that studies causality for econometrics (e.g., auctions).
- I didn't understand the sentence *The mechanised graph for Example 1 (in Figure 1b) shows that ..."* in the  the paragraph above Sec 3. Could you please clarify? Are there typos in this sentence?
- Mechanized graph is sometimes denotes by $m \mathcal G$ sometimes with $m\perp\mathcal G$. Is there any difference between the two?
- In Sec 4, what does it mean *The cyclic structure between $\Pi_{D^1}$ and $\Pi_{D^2}$ is broken,and the firm has no incentive to consider the worker’s policy*?

**Q9 Complying With Reviewing Instructions:**

Yes

---

> ### Author Rebuttal · Authors · 2024-04-08
>
> Weaknesses:
> - We agree that results related to the causal identifiability of causal effects are an interesting and important direction, though beyond the scope of the present paper. We will add a note to the future work section.
> - Computational aspects have been discussed in previous work. See Hammond et al., (2023) for other rationality assumptions - subgame perfectness & trembling hand perfectness - and see “On Imperfect Recall in Multi-agent Influence Diagrams” by Fox et al (2023) for existence guarantees of various different types of rationality assumptions as well as complexity results (we have now added a citation to this paper). The computational complexity of Algorithm 1 is (in general) intractable - but as is almost any inference problem in BNs. Fox et al (2023) discuss how decision problems in MAIDs (and causal games) such as Algorithm 1 will only be practical in restricted settings (e.g., bounded tree-width graph, agents and action sets). We leave improving the efficiency of this algorithm to future work.
> - We agree that demonstrating the method on larger examples (as well as empirically) would be interesting and important, though beyond the scope of the present paper. We have now added a note in the future work section.
>
> Detailed comments:
> - We agree about comparing this work with that in econometrics. We have now added some citations and comparisons to the related work section.
> - Thank you. There are indeed some typos here. It should say that $\Theta_T$, $\Theta_{U^1}$, and $\Pi_{D^2}$ are all $\mathcal{R}^{BR}$-relevant to $\Pi_{D^1}$.
> - Yes, the independent mechanised graph $m \perp G$ is different to the mechanised graph $mG$. The independent mechanised graph is defined as the mechanised graph without any inter-mechanism edges.
> - Thank you, this is a mistake in the graph. The graph should show that there is no longer a mutual dependency between $\Pi_1$ and $\Pi_2$.

---

### Official Review · Reviewer_X85G · 2024-03-20

**Q2-1 Originality-Novelty:** 3
**Q2-2 Correctness-Technical Quality:** 3
**Q2-5 Clarity Of Writing:** 3

**Q1 Summary And Contributions:**

This paper proposes an approach for causal games in multi-agent settings where the approach decomposes any arbitrary intervention into a set of sound and complete primitive interventions. The proposed method utilizes a mechanized graph and taxation/reward scheme to consider qualitative and quantitative specification while causal mechanism design. Finally, the method is applied to solve problems in different setups.

**Q2-3 Extent To Which Claims Are Supported By Evidence:**

2: Fair: the main claims are somewhat supported by evidence (but the experimental evaluation may be weak, or does not match entirely with the claims, important baselines may be missing, proofs contain important ideas but lack rigor, algorithmic details are only discussed superficially, references are imprecise, assumptions are not sufficiently motivated or explicated, etc.).

**Q2-4 Reproducibility:**

2: Fair: key resources (e.g. proofs, code, data) are unavailable but key details (e.g. proof sketches, experimental setup) are sufficiently well-described for an expert to confidently reproduce the main results.

**Q3 Main Strengths:**

The proposed approach seems novel and interesting. Technical contributions are discussed in a rigorous manner. The authors discussed the technical definitions in a nice and intuitive way. They nicely dealt with questions that might arise in the reader's mind while reading the paper (e.g., Remark 2). Different examples and applications also help to better understand the concepts.

**Q4 Main Weakness:**

* It is not precisely mentioned how the current work performs better than the existing work. Their novel contribution compared to existing work should be mentioned precisely.

* Although the authors discussed different interesting applications, no empirical evaluation was performed. Real-world experiments would illustrate the practical contributions of the proposed method, which is missing now.

* Some parts of the paper are a little sloppy. For example: Figure 3 uses the wrong caption. The text before Proposition 1 contains inconsistent notation.
Figure 1b is not consistent with the statement "$\theta_T$, $\theta_{U^2}$ and $\Pi_{D_1}$ are all RBR-relevant to $\Pi_{D_1}$"

**Q5 Detailed Comments To The Authors:**

Below I enlist my detailed comments and ask some questions to the authors.

* What do the dotted edges mean in Figure 1?

* The authors mentioned in the introduction that the existing methods face difficulty with the fact: an agent can observe the causal intervention
before deciding on their policy, then that agent would rationally adapt their policy in response. Why is it difficult? The authors should explain in more detail.

* Also, in the intro, the authors should provide examples of intervention from the perspective of causal games.

* Should the Def 2 be that if Y d-separates 'every' path in G from every variable in X to 'every' variable in Z?

* $\Pi_1$ and $\Pi_2$ do not take all variables as input. How do the authors justify which edges are necessary in the mechanized graph?

* The authors should discuss a little more about why the mechanism graph is a CBN.

* Can the authors explain their notation "$domain \Delta (dom(V) |dom(\mathbf{Pa}_V))$" in more detail. It is not clear from the text.

* Some examples to illustrate the completeness of the primitive interventions would be interesting to the readers.

* The decomposition into primitive interventions should be explained in more detail with examples.


* The authors mentioned that "$mG$ contains no edges of the form $M_V \rightarrow \Pi_D$". On the other hand, $\theta_T$ represents the parameterization of non-decision mechanisms and $M_V$ parameterizes the CPD.
In Figure b, there exists an edge from $\theta_T \rightarrow \Pi_{D_1}$. Thus, the first statement and the second one seem contradictory. Can the authors clear the confusion?

* Can the authors explain why taking the intersection gives the incoming edges that need to be broken? What if the intersection results in an empty set?

* How does the intervention "$do(\Pi_{D_2} = \delta(D_2, \neg j))$" imply that the firm rejects everyone?

* Figure 2b) although intervention on $D^1$ is performed but still $\Pi_{D^1}$ and $\Pi_{D_2}$ have cyclic edges. Is the cyclic edge correct?

* For incentive-invariant interventions, even though the mechanism dependency structure stays the same, shouldn't the "dependency" between the mechanism (ex: $\Pi_{D^1} \rightarrow \Pi_{D_2}$) change?

**Q9 Complying With Reviewing Instructions:**

Yes

---

> ### Author Rebuttal · Authors · 2024-04-08
>
> Weaknesses:
> - See point 2 in reply to Reviewer ZjG4
> - We agree that demonstrating the method empirically on larger examples is an interesting and important direction, though beyond the scope of this paper. We will add a note to the future work section.
> - Thank you for spotting the fact that Fig 3 uses the wrong caption and the typos. It should say that $\Theta_T$,
> $\Theta_{U^1}$, and $\Pi_{D^2}$ are all $R^{BR}$-relevant to $\Pi_{D^1}$.
>
> Detailed comments:
> - Dotted edges leading into a decision node represent the information available to the agent when the agent decides on the CPD (decision rule) for that decision node. We include these to remain consistent with previous work on causal influence diagrams (e.g., Everitt et al., 2021a and Hammond et al., 2021); however, this is actually redundant, as any parent of a decision is typically interpreted as information going into the decision. We have added this to the background section.
> - It is difficult without an explicit representation of how decision rules (policies) depend on other decision rules and parameters of the game, which is provided by the mechanised graph. 'do’ Interventions in regular causal BNs are normally simply applied to object-level nodes (cutting any edges incident to the intervened node to represent that the effect of an intervention can only propagate downstream). In a multi-agent setting, the mechanised graph is used. When all agents don’t know about the intervention (post-policy), one intervenes on an object-level node which is, by construction, never an ancestor of any decision rule node, so the pearlian `do' operation is obeyed). When all agents do know (pre-policy), then one intervenes on the decision rule node in the mechanised graph which can be upstream of another decision rule node (representing the fact that a rational agent may change their decision rule in response). This was outlined in Hammond et al (2023), but our key contribution is to extend this theory to accommodate arbitrary queries where agents choose their decision rules based on any subset of the interventions (those visible to them)
> - We have now added an example of an intervention in a causal game to the introduction. Interventions are ‘What if?’ questions that probe the potential effects of deliberate changes to the system from outside the system. For instance, `What would be the expected well-being of a worker if they were forced to pursue a university education? '
> - Yes, set $Y$ d-separates set $X$ and set $Z$ if for every possible source node $x$ in $X$ and every possible destination node $z$ in $Z$, for every path that exists between $x$ and $z$, the set of conditioned nodes $Y$ blocks that path.
> - The rationality relation $\mathcal{R}^{BR}$ defines which edges are necessary in the mechanised graph. We write there is an edge from $M_V$ into $\Pi_D$ if and only if the choice of best response decision rule $\Pi_D$ depends on the CPD at $M_V$. We have expanded on the background of this from Hammond et al. [2023].
> - Strictly the mechanised graph is now the graph of a cyclic causal model (because there may exist cycles between the decision rule nodes). A solution corresponds to a joint prob distribution consistent with all cyclic relationships, and in general, a model may have many or no solutions (see Hammond et al. [2023] for more details). We have now added more on this.
> - $\Delta(dom(V ) | dom(Pa_V ))$ means the set of all CPDs over set $V$ given the values of its parents, set $Pa_V$.
> - We agree that it is important to add examples to help illustrate the primitive interventions. We have added this.
> -  On p5 we say " independent mechanised graph $m\perp G$ which contains no edges of the form $M_V$ → $\Pi_D$". This is different to the mechanised graph $mG$. The independent mechanised graph is defined precisely as the mechanised graph without any inter-mechanism edges.
> - On the intersection point - good spot! For each reachability path, let $S_i$ be the variables with incoming edges that would break the dependency if removed. Then the minimum intervention set is not simply the intersection $X = S_1 \cap S_2 \cap … \cap S_n$. It would actually be $X$ such that $X \cap S_i \neq \emptyset$ for all i = 1, 2, ..., n. This is called the "minimum hitting set" across $\{S_1, S_2, …, S_n\}$. We have fixed this.
> - If the firm's decision rule on hiring is a Kronecker delta function with prob of 1 of choosing "not hire" ($\neg$ j), then by definition they are rejecting everyone.
> - Thank you. The cycle in figure 2b is incorrect. We have now fixed this.
> - No, the dependency doesn't change. Take two mechanism variables M1 and M2. Either they have zero reachability paths from M1 to M2 or they have at least one. This corresponds, respectively, to no arrow between M1 to M2 or an arrow from M1 to M2. By the definition of incentive invariance, the number of reachability paths stays either zero or strictly positive. Therefore the dependency between the mechanisms stays the same.

---

### Official Review · Reviewer_gKhs · 2024-03-21

**Q2-1 Originality-Novelty:** 2
**Q2-2 Correctness-Technical Quality:** 3
**Q2-5 Clarity Of Writing:** 3

**Q1 Summary And Contributions:**

The paper extends the expressivity of the recently introduced framework of causal games, offers several important theoretical results regarding this extension, and applies it to the areas of causal mechanism design and the idea of commitment.

**Q2-3 Extent To Which Claims Are Supported By Evidence:**

3: Good: the main claims are supported by convincing evidence (in the form of adequate experimental evaluation, proofs, (pseudo-)code, references, assumptions).

**Q2-4 Reproducibility:**

4: Excellent: key resources (e.g. proofs, code, data) are available and key details (e.g. proof sketches, experimental setup) are comprehensively described for competent researchers to confidently and easily reproduce the main results.

**Q3 Main Strengths:**

The paper takes a topic that is quite recent and offers novel theoretical results that contribute to their interpretation and application. The combination of causal models with game-theory offered by causal games is a sensible and exciting field, therefore getting a clearer picture of their properties is a useful contribution. Furthermore, the authors back up their theoretical results with applications that make intuitive sense.

**Q4 Main Weakness:**

The paper is very information-dense, as it consists of many definitions that build on top of each other, and thus it's quite difficult to read.

**Q5 Detailed Comments To The Authors:**

I only have two minor questions for the authors:

1: Why is there a need to consider multiple utility variables for each agent?

2: How about other decision rules than expected utility maximization? Have you looked into that?

**Q9 Complying With Reviewing Instructions:**

Yes

---

> ### Author Rebuttal · Authors · 2024-04-08
>
> Q1:
> - Causal games are a type of multi-agent influence diagram (MAID), defined by Koller and Mulch (2003). We allow each agent to have multiple utility variables to remain consistent with their definition of a MAID and Hammond et al (2023)’s definition of a causal game.
> - Multiple utility variables for each agent are permitted to express the fact that agents may have different objectives, priorities, and criteria for evaluating outcomes and one wants to make explicit this decomposition.
>
> Q2:
> - Following Hammond et al (2023), the rationality relations we refer to in this paper (denoted by $\mathcal{R}$) can capture any rationality assumptions (or decision rule). However, as we put at the end of section 2, for ease, in our examples we just assumed that agents are playing best responses  ($\mathcal{R}^{BR}$), so that the solutions of the game (i.e., where each agent is choosing their decision rules according to$\mathcal{R}^{BR}$) are the Nash equilibria of the game.
> - Still, all results in this paper are true for any rationality relation $\mathcal{R}$’ (rationality assumptions/decision rule) where a corresponding sound and complete graphical criterion $\mathcal{R}$’-reachability is known. For more on this, see Section 3 of Hammond et al. [2023].

---

### Official Review · Reviewer_ZjG4 · 2024-03-22

**Q2-1 Originality-Novelty:** 3
**Q2-2 Correctness-Technical Quality:** 3
**Q2-5 Clarity Of Writing:** 2

**Q1 Summary And Contributions:**

This paper studies the problem of causal games and characterizes a set of primitive interventions to represent any arbitrary interventional query in a multi-agent setting. The authors demonstrate the proposed method with several examples.

**Q2-3 Extent To Which Claims Are Supported By Evidence:**

2: Fair: the main claims are somewhat supported by evidence (but the experimental evaluation may be weak, or does not match entirely with the claims, important baselines may be missing, proofs contain important ideas but lack rigor, algorithmic details are only discussed superficially, references are imprecise, assumptions are not sufficiently motivated or explicated, etc.).

**Q2-4 Reproducibility:**

2: Fair: key resources (e.g. proofs, code, data) are unavailable but key details (e.g. proof sketches, experimental setup) are sufficiently well-described for an expert to confidently reproduce the main results.

**Q3 Main Strengths:**

The paper studied an interesting and relevant problem.

The proposed approach is based on a rigorous set-up of a set of primitive interventions.

The proposed method relaxes the constraints on the existing method.

**Q4 Main Weakness:**

- The paper might be a little hard for the reader who is not quite familiar with this field. (Please see comments below)

- There's not much comparison to the existing method.

- The proposed method has only been demonstrated with examples in small scales.

**Q5 Detailed Comments To The Authors:**

I feel the presentation of the paper could be improved by highlighting the challenge of the problem with existing methods, e.g., limited permissible interventions, before the technical definition and results.

Maybe it would be better to add a formal definition, such as $m\mathcal{G}$ since it is an important component used in the paper.

**Q9 Complying With Reviewing Instructions:**

Yes

---

> ### Author Rebuttal · Authors · 2024-04-08
>
> - We agree that a formal definition of the mechanised graph $mG$ would improve clarity. We have now added this.
> - We agree about restructuring the material slightly so that we expand upon the key limitations of existing methods more in the introduction. The key limitation is that existing work assumes that an intervention is either fully post-policy (entirely invisible) to all agents or fully pre-policy (entirely visible) to all agents before they decide on their decision rule at each decision point. Our most important novel contribution is to extend the theory of interventions in causal games to be able to accommodate arbitrary queries where agents choose their decision rules based on any subset of the interventions (those visible to them). This is necessary to discuss more properties of causal games (e.g. Definition 6) and then calculate certain specifications (e.g. Example 2).
> - We agree that demonstrating the method on larger examples (as well as empirically) would be interesting and important, though beyond the scope of the present paper. We have now added a note in the future work section.

---

### Meta-Review · Area_Chair_yAq1 · 2024-04-16

Previous work on causal games imposed chronological constraints on permissible interventions. This work relax this by giving a sound and complete set of primitive causal interventions so the effect of any interventional query can be studied in multi-agent settings.
I believe this results are valuable to better understanding the causality in Games and further analysing causal games.